# Two-Stage Pretraining for Molecular Property Prediction in the Wild

## Abstract

Accurate property prediction is crucial for accelerating the discovery of new molecules. Although deep learning models have achieved remarkable success, their performance often relies on large amounts of labeled data that are expensive and time-consuming to obtain. Thus, there is a growing need for models that can perform well with limited experimentally-validated data. In this work, we introduce MoleVers, a versatile pretrained model designed for various types of molecular property prediction *in the wild*, i.e., where experimentally-validated molecular property labels are scarce. MoleVers adopts a two-stage pretraining strategy. In the first stage, the model learns molecular representations from large unlabeled datasets via masked atom prediction and *dynamic denoising*, a novel task enabled by a new branching encoder architecture. In the second stage, MoleVers is further pretrained using auxiliary labels obtained with inexpensive computational methods, enabling supervised learning without the need for costly experimental data. This two-stage framework allows MoleVers to learn representations that generalize effectively across various downstream datasets. We evaluate MoleVers on a new benchmark comprising 22 molecular datasets with diverse types of properties, the majority of which contain 50 or fewer training labels reflecting real-world conditions. MoleVers achieves state-of-the-art results on 20 out of the 22 datasets, and ranks second among the remaining two, highlighting its ability to bridge the gap between data-hungry models and real-world conditions where practically-useful labels are scarce.

## 1 Introduction

A reliable molecular property prediction model enables researchers to efficiently screen vast numbers of potential compounds, reducing the need for costly experimental validation to only a select few. To this end, deep learning-based approaches have demonstrated remarkable accuracy in predicting a range of molecular properties including electronic, physical, and bioactivity properties (Yang et al., 2019; Rong et al., 2020; Fang et al., 2022; Zhou et al., 2023). However, these models typically rely on large datasets with hundreds of thousands of labeled data to achieve strong predictive performance. Yet, in real-world scenarios, labeled molecular data is often limited. For example, of the 1,644,390 assays in the ChemBL database (Zdrazil et al., 2024), only 6,113 (0.37%) contain 100 or more labeled molecules. This raises doubts about whether existing models are suitable for molecular property prediction *in the wild*, i.e., in real-world scenarios where experimentally-validated data is scarce.

In this work, we introduce MoleVers, a versatile pretrained model designed for molecular property prediction in data-scarce scenarios. MoleVers is pretrained in two stages to maximize its generalizability to various types of downstream properties. In the first stage of pretraining, we employ masked atom prediction (MAP) and dynamic denoising with relatively large noise scales to improve the generalizability of the learned representations. While previous studies have shown that increasing noise scales can impair training stability and downstream performance (Zhou et al., 2023; Yang et al., 2024), MoleVers overcomes this issue through a novel branching encoder architecture that decouples the MAP and denoising pipelines. In the second pretraining stage, MoleVers refines its representations by predicting auxiliary properties that can be derived from inexpensive computational methods such as the Density Functional Theory (DFT). Given that molecular properties are often related to molecular structures, representations learned for predicting one property can also be

useful for others. As a result of the two-stage pretraining, the model can learn molecular representations that improve the performance in downstream datasets.

To evaluate MoleVers, we introduce a new benchmark, Molecular Property Prediction in the Wild (MPPW), that consists of 22 small datasets curated from the ChemBL database (Zdrazil et al., 2024). These datasets, most of which contain 50 or fewer training labels, span a wide range of molecular properties from physical characteristics to biological activities. We standardized the pretraining datasets and data splits to ensure fair comparisons between MoleVers and several state-of-the-art pretrained models. Experimental results show that MoleVers outperforms all baselines in 20 out of the 22 assays and ranks a close second in the remaining two, while no baseline consistently ranks in the top two. Moreover, MoleVers achieves state-of-the-art performance on large datasets in the MoleculeNet benchmark (Wu et al., 2018), highlighting the effectiveness of our two-stage pretraining strategy.

In summary, our contributions are: (1) a two-stage pretraining framework that includes a novel dynamic denoising pretraining for learning molecular representations without requiring additional downstream labels, (2) a branching encoder that facilitates denoising pretraining with larger noise scales, and (3) the MPPW benchmark, designed to reflect real-world data limitations. All relevant source code will be publicly available upon publication.

## 2 RELATED WORKS

Deep learning-based molecular property prediction has demonstrated remarkable successes. Early approaches use graph neural networks (GNNs) to learn molecular representations directly from molecular structures (Kipf & Welling, 2017; Hamilton et al., 2017; Veličković et al., 2018). GNNs typically learn molecular representations by updating the node (atom) and edge (bond) features through a series of message passing across neighboring atoms. Recently, popular property prediction benchmarks such as MoleculeNet (Wu et al., 2018) are dominated by transformer-based models (Luo et al., 2022; Zhou et al., 2023; Yang et al., 2024) that leverage self-attention mechanisms to learn long-range interactions between atoms in a molecule.

Parallel to architectural advancements, pretraining has emerged as an effective strategy to improve property prediction peformance when labeled data is limited. By pretraining on a large, unlabeled dataset, a model can learn robust and transferable molecular representations that generalize well to a variety of downstream tasks. Various pretraining strategies have been proposed, including masked predictions (Wang et al., 2019; Xia et al., 2023; Zhou et al., 2023; Yang et al., 2024) and contrastive learning (Liu et al., 2022; Xia et al., 2023; Wang et al., 2022). Additionally, denoising atom coordinates and pairwise distance (Zaidi et al., 2023; Zhou et al., 2023; Liu et al., 2023) have been shown to lead to strong downstream performance. Denoising pretraining is equivalent to learning an approximate molecular force field (Zaidi et al., 2023; Liu et al., 2023), which could explain its effectiveness for improving downstream property prediction performance.

Our work is also related to the few-shot molecular property prediction. Previous studies in this area (Ju et al., 2023; Guo et al., 2021; Wang et al., 2021) often formulate the few-shot prediction as an N-way K-shot classification problem, where N classes of molecules are sampled from a dataset, each with K examples. As this formulation is not directly applicable to regression tasks, we focus our discussion in the following sections to studies that follow the pretraining-finetuning paradigm.

## 3 TWO-STAGE PRETRAINING

Our primary objective is to obtain an accurate molecular property prediction model without the need for additional, difficult-to-acquire labels for downstream tasks. To address this challenge, we propose a two-stage pretraining framework specifically designed to improve the generalization capability of our model, MoleVers. This approach enables accurate property prediction while minimizing the need for downstream labels during finetuning.

### 3.1 STAGE 1: MASKED ATOM PREDICTION AND DYNAMIC DENOISING

The properties of a molecule are strongly influenced by the spatial arrangement of its atoms in the three-dimensional (3D) space. Consequently, self-supervised pretraining that involves both atom types and 3D structures is crucial for achieving strong performance in the downstream datasets. In the first stage of our pretraining framework, we employ masked atom prediction (MAP) and dynamic denoising to train MoleVers on a large, unlabeled dataset. This encourages the model to learn representations that are transferable to downstream datasets.

#### 3.1.1 MASKED ATOM PREDICTION

Inspired by masked token prediction in natural language processing (NLP) (Devlin et al., 2019; Liu et al., 2019; Lewis et al., 2020), masked atom prediction (MAP) involves training a model to predict the correct atom types in a partially-masked molecule. This encourages the model to learn contextual relationship between atom types, capturing how they co-exist in various molecules. Multiple works (Zhou et al., 2023; Xia et al., 2023; Yang et al., 2024) have demonstrated the effectiveness of MAP as a pretraining task, which ultimately leads to better prediction models for the downstream datasets.

#### 3.1.2 DYNAMIC DENOISING

To learn information from 3D structures, we employ coordinate and pairwise distance denoising. Zaidi et al. (2023) and Liu et al. (2023) have shown that denoising tasks are equivalent to learning a molecular force field that is approximated with a mixture of Gaussians, $p(\tilde{m}) \approx q_\sigma(\tilde{m}) := \frac{1}{N} \sum_{i=1}^{N} q_\sigma(\tilde{m}|m_i)$, where $p(\tilde{m})$ is the force field, $q_\sigma(\tilde{m}|m_i) = \mathcal{N}(\tilde{m}; m_i, \sigma^2)$, and $m_1, m_2, ..., m_N$ are the equilibrium molecules in the pretraining dataset $\mathbb{D}^{\text{train}}$.

We hypothesize that using a dynamic noise scale with larger values, e.g., drawn from a uniform distribution $\sigma \sim \mathcal{U}(a, b)$, where $b > 1$, could improve the generalization ability of the model. Increasing $\sigma$ broadens each Gaussian distribution, allowing the learned force field to better cover molecules not seen in $\mathbb{D}^{\text{train}}$. Alternatively, dynamic denoising can be viewed as an augmentation technique. This follows the simple intuition that a larger $\sigma$ exposes the model to a wider set of non-equilibrium molecular configuration in a similar way to how diffusion models are trained (Ho et al., 2020; Song et al., 2021).

#### 3.1.3 DECOUPLING MAP FROM DENOISING

Unfortunately, previous works have found that setting $\sigma$ to larger values often reduces the pretraining quality and downstream performance (Zhou et al., 2023; Yang et al., 2024). This phenomenon can be explained by comparing the complexities of the MAP and denoising tasks during pretraining. In MAP, the model learns to map masked atoms ($\boldsymbol{A}^{\text{mask}}$) to their corresponding atom logits ($\hat{\boldsymbol{A}}$), $f(\boldsymbol{A}^{\text{mask}}) = \hat{\boldsymbol{A}}$, while in coordinate denoising, it learns to map noisy coordinates to their pristine values, $g(\tilde{\boldsymbol{P}}) = \hat{\boldsymbol{P}}$. The MAP function is relatively simpler because it maps a finite set of inputs (atom types) to a relatively compact set of outputs (softmax-normalized logits). In contrast, denoising deals with continuous input and output coordinates, making it more complex as the number of possible mappings is much larger. When a single model handles both MAP and denoising, the overall complexity is dominated by the more challenging denoising task. The downstream performance could then be negatively affected if the model struggles to accurately fit the complex denoising function.

This motivates us to introduce a branching encoder architecture, shown in Figure 1, that decouples the MAP and denoising pipelines. The branching design ensures that the complexity of the MAP task is minimally affected by the denoising. Furthermore, we propose to connect the two encoders with an *aggregator* module so that information can flow between the two pipelines.

#### 3.1.4 BRANCHING ENCODER

Inspired by prior works in NLP that have found masked prediction to often be the most effective pretraining tasks (Lewis et al., 2020; Raffel et al., 2020), we set the MAP encoder as the primary encoder of the model. The primary encoder, shown in Figure 1, will be passed to the second pretraining stage (Figure 2 and used for predictions in the downstream datasets.

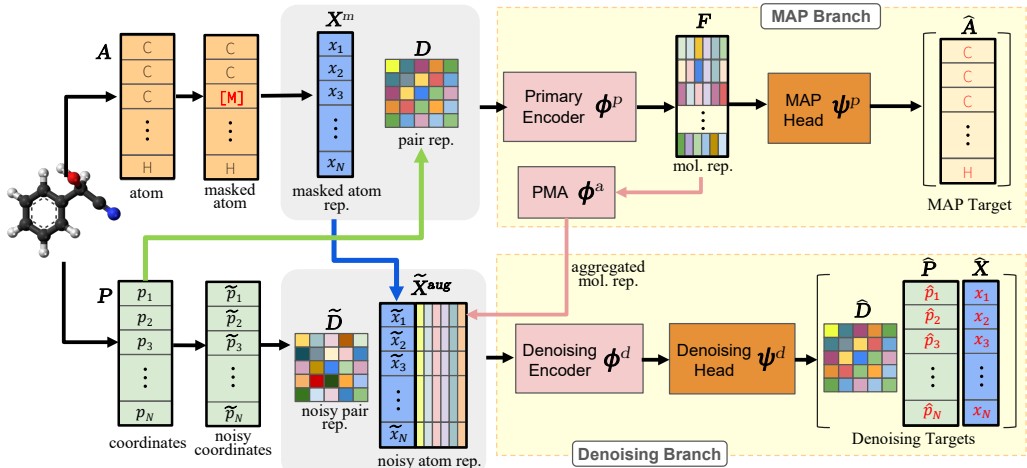

Figure 1: Illustration of pretraining **stage 1** using the proposed branching encoder. The primary encoder is assigned to the MAP branch, while another encoder with identical architecture is assigned to the denoising branch. For pretraining stage 2 and finetuning, we only keep the primary encoder and discard the denoising encoder.

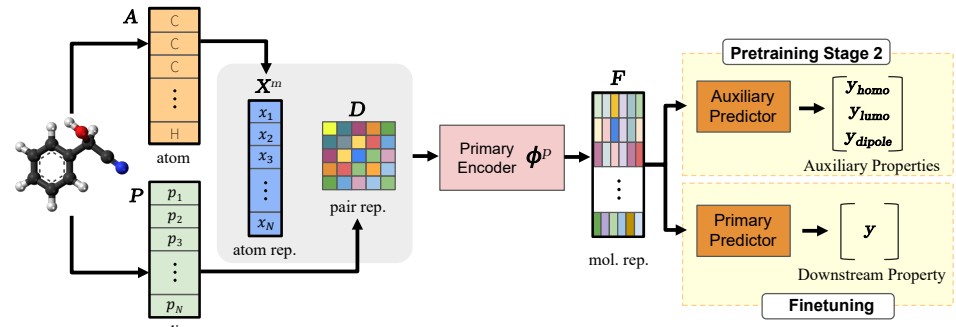

Figure 2: In pretraining **stage 2** and finetuning, we keep only the primary encoder to encode the atom and pair distance representations. A prediction head is appended to the model to predict the properties of the molecules.

The branching encoder takes as inputs the types $A \in \mathbb{Z}^N$ and coordinates $P \in \mathbb{R}^{N \times 3}$ of the $N$ atoms in a molecule. Following Zhou et al. (2023) and Yang et al. (2024), each atom type is encoded into atom representation $X \in \mathbb{R}^{N \times C}$ where $C$ is the number of features, and the coordinates are transformed into pair distance representation $D \in \mathbb{R}^{N \times N}$. During the first pretraining stage, the atom representations are masked with a ratio of $r$. We denote the masked atom representations as $X^m$.

To extract the molecule representation $F$, we feed $X^m$ and $D$ into the primary MAP encoder $\phi^p$. The logits $\hat{A}$ that represent the pristine atom types are then predicted with the MAP head $\psi^p$,

$$F = \phi^p(X^m, D), \qquad \hat{A} = \psi^p(F). \tag{1}$$

In the denoising branch, we inject noise sampled from a Gaussian distribution into $X$ and $P$ to obtain the noisy atom representations and coordinates,

$$\tilde{X} = X + \epsilon_1, \quad \tilde{P} = P + \epsilon_2, \quad \epsilon_1 \sim \mathcal{N}(\mathbf{0}, \sigma^2 I_{N \times 1}), \quad \epsilon_2 \sim \mathcal{N}(\mathbf{0}, \sigma^2 I_{N \times 3}), \quad \sigma \sim \mathcal{U}(1, 3). \tag{2}$$

To enable information flow from the denoising task to the primary MAP encoder, we augment $\tilde{\boldsymbol{X}}$ with an aggregated $\boldsymbol{F}$ using a pooling with multihead attention (PMA) module (Lee et al., 2019),

$$\tilde{\boldsymbol{X}}^{\text{aug}} = \text{concat}(\tilde{\boldsymbol{X}}, \phi^a(\boldsymbol{F})), \qquad \boldsymbol{G} = \phi^d(\tilde{\boldsymbol{X}}^{\text{aug}}, \tilde{\boldsymbol{D}}, \sigma), \qquad (\hat{\boldsymbol{X}}, \hat{\boldsymbol{P}}, \hat{\boldsymbol{D}}) = \psi^d(\boldsymbol{G}), \qquad (3)$$

where $\phi^d$ is the denoising encoder, $\phi^a$ is the PMA aggregator, $\tilde{\boldsymbol{D}}$ is derived from $\tilde{\boldsymbol{P}}$, and $\hat{\boldsymbol{X}}, \hat{\boldsymbol{P}}, \hat{\boldsymbol{D}}$ are the denoising predictions of the denoising head $\psi^d$.

### 3.2 STAGE 2: AUXILIARY PROPERTY PREDICTION

We further improve the generalization capability of the primary encoder by incorporating auxiliary property prediction in the second pretraining stage. This approach is inspired by multitask learning (Caruana, 1997), where a model is trained to solve both the primary task and related auxiliary tasks at the same time. For example, in facial analysis, the primary task might be to predict facial landmarks, while the auxiliary tasks could be to estimate head poses and infer facial attributes Zhang et al. (2014). Since these tasks share common features, the model can use the training signals from the auxiliary tasks to improve its performance in the primary task.

Given that molecular properties are heavily influenced by molecular structure, it is reasonable to assume that representations useful for predicting one type of property could also help in predicting others. Based on this intuition, we propose to construct an auxiliary dataset of properties that can be computed using relatively inexpensive computational methods, but are not necessarily identical to the properties in the downstream datasets. Specifically, we select HOMO, LUMO, and Dipole Moment as the auxiliary properties because they can be accurately computed using Density Functional Theory (DFT). We also note that computing the auxiliary labels with the DFT is cheaper than obtaining more downstream labels via real-world experiments.

In this second pretraining stage, the model is trained in a supervised manner,

$$\boldsymbol{F} = \phi^p(\boldsymbol{X}, \boldsymbol{D}), \qquad (\hat{y}_{\text{homo}}, \hat{y}_{\text{lumo}}, \hat{y}_{\text{dipole}}) = \psi^q(\boldsymbol{F}), \qquad (4)$$

where $\psi^q$ is the auxiliary predictor and $\hat{y}_{\text{homo}}, \hat{y}_{\text{lumo}}, \hat{y}_{\text{dipole}}$ are the predicted auxiliary properties. Afterward, we append the primary predictor for the downstream property to the primary encoder and finetune the model using the downstream dataset, as illustrated in Figure 2.

## 4 MOLECULAR PROPERTY PREDICTION IN THE WILD BENCHMARK

The majority of existing molecular property prediction benchmarks rely on datasets with large numbers of data points, which do not reflect real-world scenarios where such large datasets are rare. For instance, out of 1,644,390 assays available in the ChemBL database, only 6,113 assays (0.37%) contain 100 or more molecules, demonstrating the scarcity of molecular data *in the wild* where the molecular properties are validated through real-world experiments. As a result, molecular property prediction models that perform well on existing benchmark may struggle to maintain the same level of performance in real-world applications where labeled data is limited.

To address this issue, we introduce Molecular Property Prediction in the Wild (MPPW), a new benchmark specifically designed for property prediction in low-data regimes. Unlike existing benchmarks that often assume the availability of large and labeled datasets, the majority of datasets in the MPPW benchmark contain 50 or fewer training samples. This reflects the challenge faced by molecular property prediction models *in the wild*. Specifically, we have curated 22 assays from the ChemBL database (Zdrazil et al., 2024) that encompass a diverse set of properties that includes physical properties, toxicity, and biological activity. A detailed description of the datasets, including their soruces, can be found in Appendix A.1.

## 5 EXPERIMENTS AND RESULTS

In this section, we address the following questions through a series of experiments: (1) Does the two-stage pretraining framework improve the downstream performance on datasets with limited labels? (2) How does each individual pretraining stage contribute to the improvements? (3) Is our assumption that larger noise scales improve the generalization capability of the model correct? (4)

Does the choice of pretraining dataset affect downstream performance? Additionally, we investigate how significant is the impact of finetuning dataset size to the downstream performance and the results are shown in the appendix.

## 5.1 Experiment Settings

We use GDB17 (Ruddigkeit et al., 2012) as the pretraining dataset for our model *and* other models to which we compare. We randomly select 1M unlabeled molecules from the 50M subset to be used in the first pretraining stage. We then sample 130K molecules out of the 1M subset to construct the auxiliary datasets for the second pretraining stage. The labels for the auxiliary dataset are computed with Psi4 (Smith et al., 2020). We use RDKit to generate 3D conformations from SMILES (Weininger, 1988) for models that take 3D graphs as inputs.

For benchmarking purposes, we use the same pretraining dataset to minimize any performance gains that might arise from the use of higher-quality pretraining datasets. Additionally, we use identical data splits for all pretraining methods to ensure fair and consistent comparisons.

## 5.2 Implementation Details

The primary and auxiliary encoders of MoleVers are built on the UniMol encoder architecture (Zhou et al., 2023). Each encoder comprises 15 layers, with an embedding dimension of 512 and a feed-forward dimension of 2048. The MAP and denoising heads are implemented with multilayer perceptrons, while the aggregator module is implemented with pooling by multihead attention (PMA), a cross attention-based module introduced by Lee et al. (2019). The PMA module uses a *query* of size $1 \times 512$, and takes the molecule features $F$ as *key* and *value*. During the first pretraining stage, the model is trained for 1 million iterations using a batch size of 32, with a masking ratio of 0.15 for the MAP task. In the second pretraining stage, the model is trained for 50 epoch, maintaining the same batch size of 32. We employ the Adam optimizer with a learning rate of $10^{-4}$ and utilize a polynomial decay learning rate scheduler. We run all experiments on an NVIDIA Quadro RTX 8000 GPU.

## 5.3 Results on the MPPW Benchmark

In the MPPW benchmark, we compare MoleVers with four baselines: state-of-the-art GNNs, GraphMVP (Liu et al., 2022) and Mole-BERT (Xia et al., 2023), as well as state-of-the-art transformers, Uni-Mol (Zhou et al., 2023) and Mol-AE (Yang et al., 2024). All models are implemented in PyTorch (Paszke et al., 2019) and trained from scratch using publicly available source code. We also provide comparisons with more baselines on large downstream datasets in Section 5.8.

For each downstream dataset, we construct three distinct train/test splits with a 1:1 train-test ratio. All models are finetuned for 50 epochs on the training splits, and the downstream performance of the last epoch is recorded in Table 1. We evaluate the downstream performance using two metrics: mean absolute error (MAE) and the coefficient of determination ($R^2$), which indicate how well the model can explain the variance in the data.

As shown in Table 1, the $R^2$ scores for the current state-of-the-art models are relatively low. This indicates that existing models could not consistently learn molecular representations that are useful for property prediction. This highlights the need for more effective pretraining methods suited to low-data regimes. In contrast, MoleVers outperforms other baseline models in 20 out of the 22 assays, and achieving a close second rank in the remaining two. Notably, no other method consistently ranks among the top two across all assays. These results demonstrate that the two-stage pretraining framework is an effective approach for improving downstream performance when labeled data is extremely limited.

One might argue that the performance gains shown in Table 1 are simply due to the additional labels, as other methods are pretrained without auxiliary labels. To address this concern, we have conducted further experiments, detailed in Table 9 in the Appendix, where other models are also pretrained with the auxiliary labels. In such a setup too, our model achieves state-of-the-art MAE in 20 out of 22 assays, while placing a close second in the remaining two. This demonstrates that the gains achieved by MoleVers are not solely because of the additional data, but rather stem from the

Table 1: Quantitative results on the MPPW benchmark. We report the mean MAE ($\downarrow$) and $R^2$ ($\uparrow$) across three train/test splits. **Bolded** and underlined values are the best and second best results, respectively. The numbers within the parentheses after the assay ids are the number of training molecules in the assay.

| Assay | GraphMVP | | MoleBert | | UniMol | | MolAE | | MoleVers | |
|---|---|---|---|---|---|---|---|---|---|---|
| | MAE | $R^2$ | MAE | $R^2$ | MAE | $R^2$ | MAE | $R^2$ | MAE | $R^2$ |
| 1 (50) | **0.423** | **0.851** | 0.571 | 0.720 | 0.481 | 0.790 | 0.629 | 0.631 | 0.448 | 0.820 |
| 2 (50) | 0.407 | 0.261 | 0.379 | 0.317 | 0.426 | 0.148 | 0.410 | 0.183 | **0.281** | **0.582** |
| 3 (47) | 3.447 | 0.059 | 3.880 | -0.175 | 3.393 | -0.006 | 3.502 | -0.048 | **3.103** | **0.192** |
| 4 (24) | 0.329 | 0.716 | 0.591 | 0.098 | 0.475 | 0.292 | 0.590 | 0.075 | **0.319** | **0.727** |
| 5 (26) | 0.463 | -0.301 | 0.536 | -0.772 | 0.440 | -0.313 | 0.427 | -0.259 | **0.417** | **-0.246** |
| 6 (48) | 25.837 | -0.520 | 23.255 | -0.139 | 17.967 | -0.018 | 18.190 | -0.028 | **17.66** | **-0.012** |
| 7 (35) | 0.655 | 0.168 | 0.945 | -1.029 | 0.871 | -0.800 | 0.763 | -0.228 | **0.630** | **0.179** |
| 8 (30) | 0.810 | -0.829 | 0.728 | -0.371 | 0.648 | 0.067 | 0.664 | 0.002 | **0.613** | **0.106** |
| 9 (34) | 0.357 | -0.092 | 0.413 | -0.638 | 0.318 | 0.143 | 0.399 | -0.264 | **0.312** | **0.170** |
| 10 (39) | 0.233 | 0.307 | 0.258 | 0.159 | 0.211 | 0.387 | 0.271 | 0.092 | **0.187** | **0.490** |
| 11 (38) | 0.426 | 0.121 | 0.671 | -0.980 | 0.457 | 0.013 | 0.440 | 0.028 | **0.413** | **0.171** |
| 12 (25) | 0.761 | 0.066 | 0.800 | 0.190 | 0.699 | 0.233 | 0.666 | 0.403 | **0.611** | **0.412** |
| 13 (22) | 0.633 | 0.051 | 0.533 | 0.229 | 0.581 | 0.117 | 0.591 | 0.170 | **0.484** | **0.329** |
| 14 (43) | 0.351 | -0.240 | **0.267** | **0.377** | 0.331 | 0.099 | 0.303 | 0.217 | 0.280 | 0.314 |
| 15 (48) | 0.397 | 0.656 | 0.477 | 0.564 | 0.524 | 0.439 | 0.495 | 0.473 | **0.385** | **0.665** |
| 16 (24) | 0.885 | 0.279 | 0.855 | 0.287 | 0.910 | -0.080 | 0.782 | 0.301 | **0.700** | **0.364** |
| 17 (42) | 1.164 | 0.170 | 1.315 | -0.593 | 1.277 | -0.098 | 1.385 | -0.256 | **1.142** | **0.143** |
| 18 (31) | 0.195 | 0.693 | 0.170 | 0.732 | 0.202 | 0.685 | 0.202 | 0.617 | **0.141** | **0.855** |
| 19 (62) | 0.353 | -1.002 | 0.265 | -0.330 | 0.258 | -0.193 | 0.230 | -0.084 | **0.191** | **0.259** |
| 20 (51) | 0.347 | -0.557 | 0.259 | 0.260 | 0.294 | 0.010 | 0.330 | -0.217 | **0.234** | **0.343** |
| 21 (19) | 0.361 | 0.570 | 0.507 | 0.224 | 0.620 | -0.354 | 0.493 | 0.155 | **0.351** | **0.572** |
| 22 (22) | 0.983 | -0.335 | 0.733 | 0.201 | 0.608 | 0.241 | 0.580 | 0.266 | **0.526** | **0.263** |

synergy between the two pretraining stages. Additionally, we emphasize that the second pretraining stage offers a cost-effective solution for improving downstream performance, as the computational cost of obtaining auxiliary labels is negligible compared to the costs of acquiring downstream labels through wet-lab experiments.

## 5.4 ABLATION OF PRETRAINING STAGES

We study the influence of each pretraining stage on the downstream performance of MoleVers through a series of ablation studies. As shown in Table 2, incorporating either the first or second pretraining stage into the pipeline always leads to better downstream performance compared with directly training the model on the downstream datasets. Interestingly, the improvements vary across assays: some benefit more from the first pretraining stage, while others see more gains from the second pretraining stage. This variation could be due to the auxiliary properties we have chosen–HOMO, LUMO, and Dipole Moment–which are more related to intrinsic molecular properties (e.g., assay 1), rather than complex interactions (e.g., assay 3). Overall, the combination of both pretraining stages consistently yields the best downstream performance across all assays.

## 5.5 ABLATION OF BRANCHING ENCODER AND DYNAMIC DENOISING

The key components that enable denoising pretraining with higher noise levels in the first stage are the branching encoder and dynamic denoising. Here, we study the impact of each component to the downstream performance. As shown in Table 3, using a single encoder for denoising pretraining at higher noise levels generally leads to worse prediction performance. In contrast, the introduction of the branching encoder can mitigate this issue in most cases. Furthermore, combining the branching encoder with dynamic denoising consistently yields the best downstream performace, highlighting the importance of these components for the first pretraining stage.

Table 2: Ablation studies of our pretraining strategy. We report the mean MAE ($\downarrow$) and R$^2$ ($\uparrow$) across three train/test splits. We can see that combining both pretraining stage 1 and stage 2 gives the best performance on the downstream datasets.

| Pretrain Stage 1 | Pretrain Stage 2 | Assay ID | | | | | | | |
|---|---|---|---|---|---|---|---|---|---|
| | | 1 | | 2 | | 3 | | 4 | |
| | | MAE | R$^2$ | MAE | R$^2$ | MAE | R$^2$ | MAE | R$^2$ |
| - | - | 0.683 | 0.595 | 0.493 | 0.032 | 3.680 | -0.063 | 0.784 | -0.700 |
| ✓ | - | 0.592 | 0.680 | 0.420 | 0.192 | 3.161 | 0.086 | 0.431 | 0.479 |
| - | ✓ | 0.501 | 0.771 | 0.343 | 0.418 | 3.301 | 0.081 | 0.346 | 0.635 |
| ✓ | ✓ | **0.448** | **0.820** | **0.281** | **0.582** | **3.103** | **0.192** | **0.319** | **0.727** |

Table 3: Ablation studies of the proposed branching encoder and dynamic denoising. B.E. and D.D. stands for branching encoder and dynamic denoising, respectively. We report the mean MAE ($\downarrow$) and R$^2$ ($\uparrow$) across three train/test splits. Combining branching encoder with dynamic denoising yields the best downstream performance.

| B.E. | D.D. | Max $\sigma$ | Assay ID | | | | | | | |
|---|---|---|---|---|---|---|---|---|---|---|
| | | | 1 | | 2 | | 3 | | 4 | |
| | | | MAE | R$^2$ | MAE | R$^2$ | MAE | R$^2$ | MAE | R$^2$ |
| - | - | 1 | 0.481 | 0.790 | 0.426 | 0.148 | 3.393 | -0.006 | 0.475 | 0.292 |
| - | - | 10 | 0.519 | 0.769 | 0.418 | 0.135 | 3.401 | -0.044 | 0.492 | 0.377 |
| ✓ | - | 10 | 0.521 | 0.733 | 0.336 | 0.396 | 3.301 | 0.013 | 0.476 | 0.415 |
| ✓ | ✓ | 10 | 0.428 | 0.817 | 0.327 | 0.482 | 3.289 | 0.099 | 0.378 | 0.599 |

## 5.6 IMPACT OF NOISE SCALE ON DOWNSTREAM PERFORMANCE

In Section 3.1.2, we hypothesized that using larger noise scales for the denoising tasks can improve the downstream performance. In Table 4, we show the downstream performance of MoleVers with various noise scales drawn from a uniform distribution, $\sigma \sim \mathcal{U}(0, b)$, where $b$ is the maximum noise scale. Note that, similar to what has been observed in a prior work (Yang et al., 2024), the pretraining become unstable when excessively larger noise scales, e.g., $b = 20$, are used. Therefore, we limit our ablation study to a maximum value of 10.

We can see from Table 4 that, as the maximum noise scale increases, we observe consistent improvements in performance. The results confirm our hypothesis that larger noise scales could improve the downstream performance if implemented carefully. This also highlights the importance of the proposed branching encoder, which facilitates denoising pretraining with larger noise scales.

## 5.7 IMPACT OF PRETRAINING DATASET QUALITY ON DOWNSTREAM PERFORMANCE

In Section 4, we hypothesized that much of the performance gains observed in previous works may stem more from the quality of the pretraining datasets than from the pretraining method itself. Therefore, it is important to fix the pretraining dataset used in a benchmark. To test this, we examine two factors: the size of the pretraining dataset and its molecular diversity. Intuitively, a larger and more diverse set of pretraining molecules should lead to a better pretrained model compared to smaller pretraining datasets with less variation.

Table 5 shows the downstream performance of MoleVers when pretrained on datasets of varying sizes in the first stage. We observe a general trend of improved downstream performance as the pretraining dataset size increases. One exception occurs in Assay 2, where the model pretrained on 100K samples outperforms the one pretrained on 1M samples. However, the R$^2$ difference between these two models is relatively small compared to other assays, therefore, the overall trend remains valid. Furthermore, we investigate the impact of pretraining dataset diversity by filtering

Table 4: Effects of noise scales on downstream performance. We report the mean MAE ($\downarrow$) and $R^2$ ($\uparrow$) across three train/test splits. Larger noise scales tend to improve the downstream performance of MoleVers. However, using excessively large noise scales (e.g., max. $\sigma = 20$) leads to training instability.

| Max. Noise | Assay ID | | | | | | | |
|---|---|---|---|---|---|---|---|---|
| Scale $\sigma$ | 1 | | 2 | | 3 | | 4 | |
| | MAE | $R^2$ | MAE | $R^2$ | MAE | $R^2$ | MAE | $R^2$ |
| 0.1 | 0.944 | 0.193 | 0.414 | 0.251 | 3.321 | 0.042 | 0.443 | 0.496 |
| 1 | 0.658 | 0.608 | 0.464 | 0.114 | 3.486 | 0.043 | 0.559 | 0.183 |
| 3 | 0.592 | 0.680 | 0.420 | 0.192 | **3.161** | 0.086 | 0.431 | 0.479 |
| 10 | **0.428** | **0.817** | **0.327** | **0.482** | 3.289 | **0.099** | **0.378** | **0.599** |
| 20 | - | - | - | - | - | - | - | - |

Table 5: Impact of pretraining (stage 1) dataset diversity, measured by the number of training samples. We report the mean MAE ($\downarrow$) and $R^2$ ($\uparrow$) across three train/test splits. The downstream performance of MoleVers improves as the number of training samples increases.

| Dataset size | Assay ID | | | | | | | |
|---|---|---|---|---|---|---|---|---|
| (number of | 1 | | 2 | | 3 | | 4 | |
| training samples) | MAE | $R^2$ | MAE | $R^2$ | MAE | $R^2$ | MAE | $R^2$ |
| 10,000 | 1.152 | -0.008 | 0.498 | -0.083 | 3.660 | -0.047 | 0.611 | -0.016 |
| 100,000 | 0.629 | 0.636 | **0.409** | **0.198** | 3.205 | 0.103 | 0.549 | 0.189 |
| 1,000,000 | **0.592** | **0.680** | 0.420 | 0.192 | **3.161** | **0.086** | **0.431** | **0.479** |

out molecules containing specific atom types. As shown in Table 6, downstream performance generally improves as the molecular diversity of the pretraining dataset increases.

These results confirm that large and diverse pretraining datasets can improve molecular property on downstream datasets. They also highlight the importance of standardizing pretraining datasets when comparing different pretraining methods. Specifically, using the same pretraining datasets, as was done in the MPPW benchmark, ensures that any observed downstream performance improvements are the results of the pretraining strategy itself rather than variations in the pretraining dataset quality.

## 5.8 RESULTS ON LARGE DOWNSTREAM DATASETS

We further evaluate the performance of MoleVers on the MoleculeNet benchmark (Wu et al., 2018), focusing on large-scale regression datasets such as QM7, QM8, and QM9. These datasets, which range from thousands to over a hundred thousand labeled molecules, provide insights into the effectiveness of our pretraining strategy in data-abundant scenarios. As shown in Table 7, MoleVers outperforms all baseline models across all datasets, achieving the lowest MAE scores. Therefore, the proposed two-stage pretraining framework is not only effective in low-data regimes, but also excels when abundant labeled data is available.

## 6 CONCLUSION

In this work, we addressed the challenge of molecular property prediction *in the wild*, i.e., in real-world scenarios where molecular property labels that are validated through experiments are scarce. We introduced a two-stage pretraining strategy that employs masked atom prediction, dynamic denoising, and auxiliary property prediction to learn robust molecular representations. To enable effective denoising pretraining with larger noise scales, we proposed a novel branching encoder that decouples the MAP pipeline from the denoising pipeline. We evaluated our model on a new benchmark, Molecular Property Prediction in the Wild, designed to reflect real-world data limitations. Our model consistently outperforms previous state-of-the-art baselines in both low-data and high-

Table 6: Impacts of pretraining (stage 1) dataset diversity, measured by the variety of atom types. We fix the number of moelcules in each dataset to 100K for a fair comparison. We report the mean MAE ($\downarrow$) and $R^2$ ($\uparrow$) across three train/test splits. The downstream performance of MoleVers improves when the number of unique atom types in the training set increases.

| Atom Types | | | | | Assay ID | | | | | | | |
|---|---|---|---|---|---|---|---|---|---|---|---|---|
| | | | | | 1 | | 2 | | 3 | | 4 | |
| C | N | O | F | Misc. | MAE | $R^2$ | MAE | $R^2$ | MAE | $R^2$ | MAE | $R^2$ |
| ✓ | ✓ | - | - | - | 1.093 | 0.018 | 0.496 | -0.052 | 3.584 | -0.013 | 0.628 | -0.015 |
| ✓ | ✓ | ✓ | - | - | 0.845 | 0.426 | 0.431 | 0.191 | 3.428 | 0.041 | 0.480 | 0.416 |
| ✓ | ✓ | ✓ | ✓ | - | 0.619 | 0.601 | 0.423 | **0.233** | 3.273 | 0.055 | 0.493 | 0.281 |
| ✓ | ✓ | ✓ | ✓ | ✓ | **0.592** | **0.680** | **0.420** | 0.192 | **3.161** | **0.086** | **0.431** | **0.479** |

Table 7: Results on larger datasets. We use three large regression datasets of the MolculeNet benchmark: QM7, QM8, and QM9. The MAE values of methods other than MoleVers are obtained from Yang et al. (2024).

| Dataset | QM7 | QM8 | QM9 |
|---|---|---|---|
| #Molecules | 6830 | 21789 | 133885 |
| D-MPNN | 103.5 | 0.0190 | 0.0081 |
| Attentive FP | 72.0 | 0.0179 | 0.0081 |
| Pretrain-GNN | 113.2 | 0.0200 | 0.0092 |
| GROVER | 94.5 | 0.0218 | 0.0099 |
| MolCLR | 66.8 | 0.0178 | - |
| Uni-Mol | 58.9 | 0.0160 | 0.0054 |
| Mol-AE | 53.8 | 0.0161 | 0.0053 |
| MoleVers (ours) | **51.3** | **0.0155** | **0.0050** |

data regimes. Our results highlight the effectiveness of the two-stage pretraining strategy, making it suitable for real-world applications where labeled data are extremely limited.

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

## A  APPENDIX

### A.1  DETAILS OF DATASETS USED IN THE MPPW BENCHMARK

The Molecular Property Prediction in the Wild (MPPW) benchmark uses two types of datasets: pretraining datasets and downstream datasets. For our first-stage pretraining, as well as in the pretraining of other models shown in Table 1, we randomly select 1M unlabeled molecules from the GDB17 dataset (Ruddigkeit et al., 2012). For the second-stage pretraining, we sample around 130K molecules from the 1M subset and calculate the auxiliary labels—HOMO, LUMO, and Dipole Moment—using Psi4 (Smith et al., 2020). This smaller subset is also used to pretrain other models shown in 9.

For downstream evaluation, we curated 22 small datasets from the ChemBL database (Zdrazil et al., 2024), representing a diverse set of molecular properties as detailed in Table 8. To ensure consistency across datasets, we filter out any molecules containing atoms not present in the GDB17 dataset. As a result, only molecules containing the atoms {H, C, N, O, S, F, Cl, Br, I} are included in the downstream datasets. For evaluation, each dataset is randomly sampled to create three train/test splits with a 50:50 ratio, and all models in Tables 1 and 9 are assessed using these same splits. The processed datasets can be accessed through this URL.

### A.2  MORE RESULTS ON THE MPPW BENCHMARK

As an additional experiment, we evaluated the downstream performance of MoleVers alongside the baseline models. In this experiment, the baselines are first pretrained using their original pretraining strategy, followed by a second-stage pretraining via auxiliary property prediction. The results, presented in Table 9, show that MoleVers achieves state-of-the-art MAE in 20 out of the 22 datasets, and ranks second in the remaining two. In terms of $R_2$ scores, MoleVers achieves the best performance in 19 out of the 22 datasets, and ranks second in the other three. Note that none of the other models consistently rank in the top two across all datasets. Since all models are pretrained with the auxiliary labels, the results in Table 9 further highlight the benefits of our branching encoder which enables denoising pretraining with larger noise scales.

### A.3  IMPACT OF FINETUNING DATASET SIZE ON DOWNSTREAM PERFORMANCE

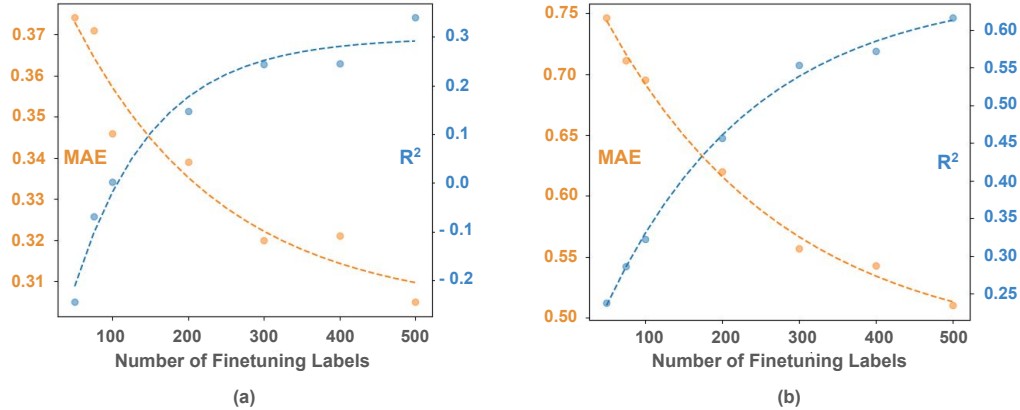

Figure 3: Predictive performance of MoleVers, averaged over 5 splits, when finetuned on two assays with varying dataset size: (a) CHEMBL5291763, (b) CHEMBL2328568 (Zdrazil et al., 2024).

To assess the impact of finetuning dataset size on downstream performance, we gradually reduce the number of training labels used to finetune MoleVers, and validate it on fixed validation sets. We conduct this experiment using two large datasets outside the MPPW benchmark, as the datasets in the benchmark contain only a limited number of molecules. As shown in Figure 3, the MAE curves show exponential decay as the number of finetuning labels increases, while the $R^2$ curves exhibit logarithmic growth. This demonstrates a sharp drop in prediction quality, especially when

Table 8: Details for datasets in the MPPW benchmark. We curated 22 small datasets of diverse properties from the ChemBL database. The last two datasets are used for ablation (Section A.3).

| ID | ChemBL ID | #Mols. | Short Description | Unit | Reference |
|---|---|---|---|---|---|
| 1 | 635482 | 100 | Partition coefficient (logP) | - | Hansch et al. (1980) |
| 2 | 4150258 | 99 | Antimycobacterial activity against Mycobacterium bovis BCG ATCC 35734 | log nM | Nyantakyi et al. (2018) |
| 3 | 744489 | 94 | Antimalarial activity in Plasmodium berghei infected mice (Mus musculus) | - | Kesten et al. (1992) |
| 4 | 638473 | 48 | Partition coefficient (logD7.4) | - | Rai et al. (1998) |
| 5 | 5251479 | 51 | Induction of mitochondrial uncoupling activity in rat L6 cells assessed as increase in oxygen consumption rate | log nM | Murray et al. (2023) |
| 6 | 778368 | 95 | Hypolipidemic effects(plasma TG) in male rats | % | Sircar et al. (1983) |
| 7 | 813331 | 69 | Inhibitory activity against Tachykinin receptor 1 | log nM | Vedani et al. (2000) |
| 8 | 3375151 | 60 | Antimycobacterial activity against Mycobacterium kansasii CNCTC My 235/80 | log nM | Karabanovich et al. (2014) |
| 9 | 687437 | 68 | Bronchodilator activity against histamine- induced spasm in guinea pig | log umol kg$^{-1}$ | Hermecz et al. (1987) |
| 10 | 4770530 | 78 | Cytotoxicity against human TZM-GFP cells | log nM | Wang et al. (2020) |
| 11 | 3282634 | 75 | Antitumor activity against mouse L1210 cells transfected in ip dosed C3H/DBA2 F1 mouse qd | log mg kg$^{-1}$ day$^{-1}$ | Denny et al. (1978) |
| 12 | 632430 | 50 | Partition coefficient (logP) (chloroform) | - | Dunn III et al. (1987) |
| 13 | 950577 | 44 | Antifungal activity against Candida albicans | - | Katritzky et al. (2008) |
| 14 | 984427 | 85 | Antiviral activity against CVB2 infected in Vero76 cells | log nM | Tonelli et al. (2008) |
| 15 | 1862759 | 96 | DNDI: Lipophilicity measured in Chromatographic hydrophobicity index assay, pH 7.4 | - | |
| 16 | 3066822 | 47 | Dissociation constant, pKa of the compound at pH 7.3 | - | Akamatsu (2011) |
| 17 | 3745095 | 84 | Antifungal activity against Candida glabrata clinical isolate | log$_2$ ug ml$^{-1}$ | De Monte et al. (2016) |
| 18 | 4835984 | 61 | Brain to blood partition coefficient of the compound | - | Li et al. (2021) |
| 19 | 4888494 | 123 | Re-testing in dose-response curve in HepG2 cytotoxicity assay, at 72h | log nM | Dechering et al. (2022) |
| 20 | 5043600 | 101 | Cytotoxicity in dog MDCK cells assessed as reduction in cell viability | log nM | Mizuta et al. (2021) |
| 21 | 1070367 | 38 | ABTS radical scavenging activity assessed as trolox equivalent antioxidant capacity | log MU | Amić & Lučić (2010) |
| 22 | 2427705 | 44 | Half life in phosphate buffer at pH 7.4 at 50 uM | log hour | Ward et al. (2013) |
| A | 5291763 | 1237 | Inhibition of NaV1.7 ion channel | log nM | (Sutherland et al., 2023) |
| B | 2328568 | 1017 | Inhibition of human CHRM1 | log nM | (Norinder & Ek, 2013) |

the number of finetuning labels fall below 200. These results emphasize the inherent challenge of molecular property prediction *in the wild* due to the scarcity of labeled data in real-world. The observed performance degradation with smaller datasets also highlights the importance of an effective pretraining strategy, such as the proposed two-stage pretraining approach of MoleVers, in mitigating the limitations imposed by limited labeled data.

Table 9: Quantitative results on the MPPW benchmark. The ++ version of existing methods are trained in two stages: first with their vanilla pretraining strategy, then with auxiliary predictions. We report the mean MAE ($\downarrow$) and $R^2$ ($\uparrow$) across three train/test splits. **Bolded** and underlined values are the best and second best results, respectively. The numbers within the parentheses after the assay ids are the number of training molecules in the assay.

| Assay | GraphMVP++ | | MoleBert++ | | UniMol++ | | MolAE++ | | MoleVers | |
| --- | --- | --- | --- | --- | --- | --- | --- | --- | --- | --- |
| | MAE | $R^2$ | MAE | $R^2$ | MAE | $R^2$ | MAE | $R^2$ | MAE | $R^2$ |
| 1 (50) | 0.482 | 0.803 | 0.567 | 0.720 | 0.460 | 0.806 | **0.446** | 0.759 | 0.448 | **0.820** |
| 2 (50) | 0.375 | 0.386 | 0.297 | 0.524 | 0.303 | 0.534 | 0.356 | 0.306 | **0.281** | **0.582** |
| 3 (47) | 3.512 | -0.054 | 3.373 | 0.017 | 3.397 | 0.063 | 3.231 | 0.109 | **3.103** | **0.192** |
| 4 (24) | 0.332 | 0.673 | 0.499 | 0.200 | 0.383 | 0.590 | 0.358 | 0.641 | **0.319** | **0.727** |
| 5 (26) | 0.447 | -0.263 | 0.590 | -1.322 | 0.430 | -0.377 | 0.431 | **-0.176** | **0.417** | -0.246 |
| 6 (48) | 26.624 | -0.540 | 28.197 | -0.494 | 18.54 | -0.028 | 17.85 | -0.013 | **17.66** | **-0.012** |
| 7 (35) | 0.697 | 0.031 | 0.887 | -0.493 | 0.723 | -0.054 | 0.748 | -0.179 | **0.630** | **0.179** |
| 8 (30) | 0.645 | -0.147 | 0.803 | -0.829 | 0.687 | -0.279 | 0.613 | 0.070 | **0.613** | **0.106** |
| 9 (34) | 0.336 | -0.250 | 0.330 | 0.020 | 0.319 | 0.107 | 0.330 | 0.090 | **0.312** | **0.170** |
| 10 (39) | 0.218 | 0.335 | 0.196 | 0.348 | 0.226 | 0.313 | 0.220 | 0.348 | **0.187** | **0.490** |
| 11 (38) | 0.426 | 0.043 | 0.451 | 0.020 | 0.448 | 0.050 | 0.432 | 0.049 | **0.413** | **0.171** |
| 12 (25) | 0.738 | 0.251 | 0.738 | 0.256 | 0.653 | 0.337 | 0.653 | 0.327 | **0.611** | **0.412** |
| 13 (22) | 0.627 | 0.089 | 0.627 | -0.120 | 0.640 | -0.125 | 0.604 | 0.053 | **0.484** | **0.329** |
| 14 (43) | 0.343 | -0.321 | **0.259** | **0.405** | 0.373 | -1.119 | 0.310 | 0.084 | 0.280 | 0.314 |
| 15 (48) | 0.426 | 0.619 | 0.441 | 0.547 | 0.572 | 0.199 | 0.467 | 0.519 | **0.385** | **0.665** |
| 16 (24) | 0.928 | 0.124 | 0.719 | 0.360 | 0.838 | 0.231 | 0.839 | 0.076 | **0.700** | **0.364** |
| 17 (42) | 1.186 | 0.141 | 1.284 | -0.230 | 1.244 | -0.062 | 1.274 | -0.087 | **1.142** | **0.143** |
| 18 (31) | 0.188 | 0.692 | 0.188 | 0.721 | 0.165 | 0.761 | 0.149 | 0.815 | **0.141** | **0.855** |
| 19 (62) | 0.278 | -0.360 | 0.250 | -0.215 | 0.227 | -0.126 | 0.192 | 0.160 | **0.191** | **0.259** |
| 20 (51) | 0.289 | -0.043 | 0.257 | 0.202 | 0.241 | 0.158 | 0.277 | 0.052 | **0.234** | **0.343** |
| 21 (19) | 0.383 | 0.525 | 0.486 | 0.222 | 0.431 | 0.322 | 0.376 | 0.518 | **0.351** | **0.572** |
| 22 (22) | 0.671 | 0.016 | 0.641 | 0.103 | 0.643 | 0.130 | 0.556 | **0.314** | **0.526** | 0.263 |

