# OpenReview forum: "Two-Stage Pretraining for Molecular Property Prediction in the Wild"
_ICLR.cc/2025/Conference — Submitted to ICLR 2025_

### Official Review · Reviewer_StiN · 2024-10-31

**Soundness:** 2
**Presentation:** 2
**Contribution:** 1
**Rating:** 3
**Confidence:** 5

**Summary:**

This paper introduces MoleVers, a two-stage pre-training framework designed for molecular property prediction. The first stage of pre-training includes a masking atom prediction strategy alongside a dynamic denoising approach. The second stage involves supervised learning tasks, leveraging labels generated by DFT methods.

**Strengths:**

1. The paper is well-written and easy to follow.
2. The dynamic scaling of the denoising process is an innovative and interesting approach.

**Weaknesses:**

1. Overall, the proposed pre-training strategy lacks novelty. Both the masking and denoising strategies have been widely adopted in previous pre-training methods, such as Uni-Mol[1] and 3D-denoising. Additionally, the second stage of pre-training, which involves supervised task learning with DFT-generated properties, is also used by Transformer-M[2]. The motivation for splitting the pre-training into two distinct stages needs further clarification.
[1] Uni-Mol: A Universal 3D Molecular Representation Learning Framework
[2] One Transformer Can Understand Both 2D & 3D Molecular Data

2. The DFT process is computationally expensive, especially for large molecules. Relying on DFT-generated labels for pre-training tasks in MRL may not be a feasible or efficient choice.
3. The rationale for decoupling the denoising and masking atom tasks is unclear, as these two distributions operate on different modalities—coordinates and atom types, respectively.
4. Results for each subtask on QM9 and MoleculeNet should be reported to demonstrate the model's superiority effectively.

**Questions:**

See weakness

---

> ### Author Response · Authors · 2024-11-19
>
> **The proposed pre-training strategy is not novel enough as the masking and denoising strategies have been widely adopted in previous pre-training methods, such as Uni-Mol [1] and 3D-denoising.**
>
> We are in full agreement that masked atom prediction (MAP) and denoising tasks, both used in our first pretraining stage, have been widely adopted in various works. However, the technical novelty of our first pretraining stage lies in the dynamic denoising and the branching encoder, which decouples the denoising and MAP tasks. This novel design fixes a significant problem in denoising pretraining, where it was believed that large noise scales (e.g., those leading to larger than **1 Å** atom displacement) typically hurt the downstream performance [1-4]. With our model, we demonstrate that using larger noise scales (up to **10 Å**) actually improves the downstream performance—a new insight for denoising-based molecular pretraining.
>
> We also note that our technical contributions go beyond the pretraining strategy. Specifically, we:
> - Propose a new benchmark (Section 4) that reflects the conditions in the real-world. Here, the datasets contain 50 or less training samples with experimentally-validated properties. We believe our benchmark is more realistic compared with QMs or MoleculeNet, which use datasets with hundreds to hundreds of thousand training samples. For reference, only around 0.37% of assays in the ChemBL database have 100 or more labeled molecules.
> - Standardize the pretraining dataset in our benchmark to limit the influence of pretraining dataset quality to the downstream performance. The standardization is important to ensure fairness when comparing multiple pretraining methods, as pretraining dataset quality can significantly affect the downstream performance (Section 5.6)
> - Provide extensive discussions on the effects of (1) noise scales, (2) pretraining dataset size, (3) pretraining dataset diversity.
>
> **The rationale for decoupling the denoising and masking atom tasks is unclear, as these two distributions operate on different modalities—coordinates and atom types, respectively.**
>
> We are happy to expand and improve the rationale for the decoupling, which is already discussed in Section 3.1.3. (line 142 - 152). The reviewer is right that these tasks operate on different modalities (coordinates and atom types). However, their joint learning via a shared encoder is suboptimal as the network capacity would be disproportionately allocated to solving the more complex pretraining task, denoising, at the expense of the MAP performance. By introducing a branching structure and connecting the branches through an aggregation module, we allow each task-specific encoder to learn independently and focus on its respective objective. This separation reduces interference between the tasks and enables more efficient optimization, ultimately leading to better downstream performance.
>
> **Supervised task learning with DFT-generated properties is also used by Transformer-M[5]. The motivation for splitting the pre-training into two distinct stages needs further clarification.**
>
> A single-stage approach such as used by Transformer-M [5] would require either (a) labeling all molecules in the dataset, or (b) a bespoke model whose components can be dynamically activated or deactivated according to label availability. In contrast, our two-stage pretraining approach offers a practical advantage: we can use a large, unlabeled dataset in the first stage and a smaller, labeled dataset in the second stage in a plug and play manner. Any models that can learn via self-supervision can be trained with our approach with minimal modifications (Appendix A2).
>
> **The DFT process is computationally expensive, especially for large molecules. Relying on DFT-generated labels for pre-training tasks in MRL may not be a feasible or efficient choice.**
>
> Calculating the HOMO, LUMO, and dipole moment of 100K molecules available in GDB17 only takes less than 48 hours. It is a one-time computation that can improve the prediction performance at a negligible cost compared with collecting new labels through wet lab-based experiments (Line 352-355).

---

> > ### Author Response · Authors · 2024-11-19
> >
> > **Results for each subtask on QM9 and MoleculeNet should be reported to demonstrate the model's superiority effectively.**
> >
> > Thank you for the suggestion! We have conducted additional experiments on regression tasks in the MoleculeNet benchmark (high-data regime), and the results are shown in the table below. Combined with results in Table 7, we can see that our method outperforms state-of-the-art models in high-data regimes. Note that we have trained the baselines from scratch using the GDB17 dataset. This is to ensure that any performance gains are the results of the pretraining strategy, not the pretraining dataset quality.
> >
> > It is important to note that our work primarily focuses on real-world scenarios where labeled training data often consists of 50 molecules or fewer. Therefore, the results on our MPPW benchmark (small-data regime) should be considered as the core evaluation, while results from QMs and MoleculeNet are included only for supplementary comparisons.
> >
> > Table A. Results on MoleculeNet (regression tasks). We report the RMSE of each method.
> > | Dataset   | Ours (pretrained with GDB17) | UniMol (pretrained with GDB17) | Mol-AE (pretrained with GDB17) |
> > |-----------|-------------------------------|--------------------------------|--------------------------------|
> > | ESOL      | 0.8036                        | 0.8497                         | 0.9524                         |
> > | LIPO      | 0.6354                        | 0.6465                         | 0.6965                         |
> > | Freesolv  | 1.5944                        | 1.5769                         | 2.5861                         |

---

> ### Comment · Reviewer_StiN · 2024-11-24
>
> Thanks for authors' responses. Some of my concerns have been addressed. However, I still feel that the pre-training strategies lack novelty, and the motivation behind decoupling denoising and MAP remains unconvincing to me. I will discuss with the other reviewers before reaching a final decision.

---

### Official Review · Reviewer_BLw3 · 2024-11-04

**Soundness:** 3
**Presentation:** 3
**Contribution:** 3
**Rating:** 5
**Confidence:** 4

**Summary:**

In real-world scenarios where molecular property labels are scarce, there is a need to develop property prediction methods that can perform well with limited labeled data. To address this, the authors propose a two-stage pretraining approach, MoleVers. The first stage employs atom masking and dynamic denoising, where a branching encoder is used to mitigate interference between denoising pretraining with larger noise scales and the masking task. To evaluate MoleVers' fine-tuning performance on limited labeled data, the authors introduce the MPPW benchmark, comprising 22 biological and chemical property subsets, each containing between 38 and 123 molecules. MoleVers outperforms four baseline molecular pretraining methods and also shows superior performance on QM property prediction datasets with abundant data. The ablation studies demonstrate the effectiveness of both pretraining stages, as well as the impact of noise scale and pretraining data on performance.

**Strengths:**

1. The writing is clear and easy to understand.
2. Experimental settings are well-detailed, with fair comparisons based on a shared pretraining dataset, making the results convincing.
3. Addressing the challenge of limited labeled data for biological and chemical property prediction is crucial, and the two-stage approach provides an effective solution that could inspire further research in this area.

**Weaknesses:**

1. Given that each task in the proposed MPPW benchmark includes only 38–123 molecules, there is a risk that the results are not robust and variable. However, I did not see any mention of repeated experiments or the reporting of standard deviations. I suggest the authors perform multiple runs and report the mean and standard deviation to make the results in Table 1 more robust and convincing.
2. The authors propose a dynamic denoising pretraining strategy and a branching network design, but no ablations are provided to test the effectiveness of these design choices. Could the authors conduct experiments to compare dynamic and static denoising pretraining, as well as examine the impact of branching on the results?
3. The model descriptions could be more thorough to aid reader comprehension. For example, what is the network structure of the Denoising and MAP heads? What is the "pooling with multihead attention" (PMA) module, and what does the pair representation entail? A detailed description of the encoder architecture would also be helpful, perhaps included in the main text or the appendix.

**Questions:**

1. In Section 3.1, you mention that "the denoising tasks are equivalent to learning a molecular force field" and that this explanation is based on the assumption that the input data is at equilibrium. Do the RDKit-generated conformations used here meet this "equilibrium" assumption?
2. In Section 3.1.2, you mention that increasing the noise scale can cover more unseen molecules. This issue was also addressed in [1], where they argued that the noise scale of an isotropic Gaussian correlates with the accuracy of force fields, showing that excessively small or large noise scales negatively impact force field accuracy (extended data Fig. 3 [1]). In contrast, this paper suggests that larger noise scales are beneficial. How do you think of this discrepancy?
[1] Pre-training with fractional denoising to enhance molecular property prediction, Nature Machine Intelligence, 2024.
3. Additionally, in the dynamic denoising pretraining, which is compared to the denoising diffusion model, does the dynamic noise scale $$\sigma$$ serve as an input to the network as in diffusion models?
4. In Section 3.2, you selected HOMO, LUMO, and dipole moment prediction as the second-stage pretraining tasks. Why were these tasks chosen? Are they related to downstream properties in MPPW? Regarding your assumption that "representations useful for predicting one type of property could also help in predicting others," does this require that the second-stage properties are correlated with downstream properties? After all, property prediction tasks are different from general pre-training tasks and are already learning high-level features.

---

> ### Author Response · Authors · 2024-11-19
>
> **I did not see any mention of repeated experiments or the reporting of standard deviations. I suggest the authors perform multiple runs and report the mean and standard deviation to make the results in Table 1 more robust and convincing.**
>
> All experimental results on our benchmark in the Table (1-6, 8) are the average of three different train/test splits (Section 5.3). Unfortunately, we could not include the standard deviations without removing one baseline due to the space limitation. We provide a snippet of the prediction performance on the individual splits in Table A below.
>
> **Could the authors conduct experiments to compare dynamic and static denoising pretraining, as well as examine the impact of branching on the results?**
>
> Thank you for the suggestion! We have conducted an additional ablation study focusing on the effects of the branching encoder and dynamic noise. The results, presented in Table B below, show a general trend where downstream performance degrades with higher noise levels when a single encoder with static noise approach is used. In most cases, utilizing the branching encoder resolves this issue. Ultimately, combining the branching encoder with dynamic noise yields the best performance across all four assays. We will include this in the revised manuscript.
>
> **What is the network structure of the Denoising and MAP heads? What is the "pooling with multihead attention" (PMA)**
>
> The MAP and denoising heads are implemented with multilayer perceptrons, while the aggregator module is implemented with pooling by multihead attention (PMA), a cross attention-based module introduced in [10]. The PMA module uses a query of size 1 x 512, and takes the molecule features F as key and value. We have added this explanation to the revised manuscript.
>
> **Do the RDKit-generated conformations used here meet this "equilibrium" assumption?**
> RDKit-generated conformations are not necessarily at equilibrium and are less accurate than DFT calculations. However, they are sufficiently accurate for the denoising pretraining task, as also noted by [4].
>
> **[4] argued that excessively small or large noise scales negatively impact force field accuracy (extended data Fig. 3 [4]). In contrast, this paper suggests that larger noise scales are beneficial. How do you think of this discrepancy?**
>
> Thank you for bringing this work to our attention! We missed this work as it was published shortly before the submission deadline. We believe that both conclusions can be true, since the network architectures are different. The hypothesis in [4] is that “larger noise scales yield more irrational noisy samples” that could “degenerate the (downstream) performance notably”. In [4], the pretraining utilizes static noise scales and does not provide the noise scale information to the encoder. In our work, the denoising encoder is conditioned on the dynamic noise scale, i.e., the network is now aware of how noisy the input is, and is learning to denoise a wide variety of noisy inputs, from trivial to challenging noises. Therefore, the noise scale information and the trivial noisy samples can help the network “rationalize” the more irrational, high-noise samples.
>
> **Additionally, in the dynamic denoising pretraining, which is compared to the denoising diffusion model, does the dynamic noise scale σ serve as an input to the network as in diffusion models?**
>
> Yes, the noise scale serves as an input to the network. We will fix Figure 1 and Eq. 3 to better reflect this. Thank you for raising this question.
>
> **In Section 3.2, you selected HOMO, LUMO, and dipole moment prediction as the second-stage pretraining tasks. Why were these tasks chosen? Regarding your assumption that "representations useful for predicting one type of property could also help in predicting others," does this require that the second-stage properties are correlated with downstream properties?**
>
> Our hypothesis is based on the fact that all molecular properties are a function of their 3D structures (Line 232). Therefore, the pretraining properties do not have to be strongly-correlated with the downstream properties for the supervised pretraining to be beneficial. This is demonstrated in Table 1, as our model achieves good performance across the diverse type of properties in MPPW. We choose HOMO, LUMO, and dipole moment for the pretraining properties because they can be calculated within reasonable time and accuracy using DFT.

---

> > ### Author Response · Authors · 2024-11-19
> >
> > Table A. Quantitative results on the individual splits of the MPPW benchmark.
> >
> > | Assay Idx | Split | Ours  |       | UniMol |        | GraphMVP |       | MoleBert |        | Mol-AE |        |
> > |-----------|-------|-------|-------|--------|--------|----------|-------|----------|--------|--------|--------|
> > |           |       | MAE   | R2    | MAE    | R2     | MAE      | R2    | MAE      | R2     | MAE    | R2     |
> > | 1         | 1     | 0.437 | 0.810 |  0.535 |  0.741 |    0.439 | 0.846 |    0.656 |  0.660 |  0.801 |  0.438 |
> > |           | 2     | 0.424 | 0.839 |  0.399 |  0.842 |    0.409 | 0.849 |    0.524 |  0.765 |  0.450 |  0.744 |
> > |           | 3     | 0.482 | 0.809 |  0.510 |  0.788 |    0.420 | 0.858 |    0.532 |  0.734 |  0.635 |  0.711 |
> > |           | Mean  | 0.448 | 0.820 |  0.481 |  0.790 |    0.423 | 0.851 |    0.571 |  0.720 |  0.629 |  0.631 |
> > | 2         | 1     | 0.297 | 0.530 |  0.331 |  0.428 |    0.386 | 0.317 |    0.408 |  0.239 |  0.354 |  0.282 |
> > |           | 2     | 0.270 | 0.591 |  0.458 |  0.171 |    0.400 | 0.309 |    0.347 |  0.427 |  0.396 |  0.300 |
> > |           | 3     | 0.277 | 0.624 |  0.490 | -0.156 |    0.435 | 0.158 |    0.384 |  0.284 |  0.481 | -0.031 |
> > |           | Mean  | 0.281 | 0.582 |  0.426 |  0.148 |    0.407 | 0.261 |    0.379 |  0.317 |  0.410 |  0.183 |
> > | 3         |     0 | 2.471 | 0.315 |  2.885 | -0.043 |    3.002 | 0.113 |    3.784 | -0.389 |  3.108 | -0.115 |
> > |           |     1 | 3.166 | 0.144 |  3.302 |  0.066 |    3.482 | 0.060 |    3.351 |  0.121 |  3.447 | -0.023 |
> > |           |     2 | 3.673 | 0.118 |  3.993 | -0.041 |    3.858 | 0.004 |    4.505 | -0.257 |  3.951 | -0.006 |
> > |           | Mean  | 3.103 | 0.192 |  3.393 | -0.006 |    3.447 | 0.059 |    3.880 | -0.175 |  3.502 | -0.048 |
> > | 4         |     0 | 0.399 | 0.621 |  0.563 |  0.050 |    0.416 | 0.601 |    0.691 | -0.089 |  0.565 |  0.073 |
> > |           |     1 | 0.287 | 0.740 |  0.474 |  0.125 |    0.320 | 0.689 |    0.572 | -0.001 |  0.590 |  0.020 |
> > |           |     2 | 0.273 | 0.819 |  0.387 |  0.702 |    0.252 | 0.859 |    0.511 |  0.385 |  0.615 |  0.133 |
> > |           | Mean  | 0.319 | 0.727 |  0.475 |  0.292 |    0.329 | 0.716 |    0.591 |  0.098 |  0.590 |  0.075 |
> >
> >
> > Table B. Ablation studies of the proposed branching encoder and dynamic denoising.
> > | Branching | Dynamic Noise | Max Noise Level | Assay 1 |       | Assay 2 |       | Assay 3 |        | Assay 4 |       |   |
> > |-----------|---------------|-----------------|---------|-------|---------|-------|---------|--------|---------|-------|---|
> > |           |               |                 | MAE     | R2    | MAE     | R2    | MAE     | R2     | MAE     | R2    |   |
> > | x         | x             | 1               |   0.481 | 0.790 |   0.426 | 0.148 |   3.393 | -0.006 |   0.475 | 0.292 |   |
> > | x         | x             | 10              |   0.519 | 0.769 |   0.418 | 0.135 |   3.401 | -0.044 |   0.492 | 0.377 |   |
> > | v         | x             | 10              |   0.521 | 0.733 |   0.336 | 0.396 |   3.301 |  0.013 |   0.476 | 0.415 |   |
> > | v         | v             | 10              |   0.428 | 0.817 |   0.327 | 0.482 |   3.289 |  0.099 |   0.378 | 0.599 |   |

---

> > > ### Comment · Reviewer_BLw3 · 2024-11-26
> > >
> > > Thank you for your response and additional experiments. Some of my concerns have been addressed.
> > >
> > > However, several critical issues remain unsolved, like:
> > >
> > > 1. comparisons between  dynamic and static denoising pretraining, as well as the impact of branching
> > > 2. ablation of different network structure of the Denoising and MAP heads
> > > 3. theoretical and implementation gap with the "equilibrium" assumption
> > > 4. more explanation on the dynamic noise scale σ
> > >
> > > Therefore, I will maintain my score unchanged.

---

### Official Review · Reviewer_kQgY · 2024-11-04

**Soundness:** 3
**Presentation:** 3
**Contribution:** 2
**Rating:** 5
**Confidence:** 4

**Summary:**

The paper introduces MoleVers, a two-stage pretrained model aimed at improving molecular property prediction in low-data scenarios. The model first undergoes self-supervised training through masked atom prediction (MAP) and dynamic denoising. In the second stage, MoleVers utilizes auxiliary labels generated by cost-effective computational methods to further refine molecular representations. The model is evaluated on a newly curated benchmark, which includes 22 datasets with 50 or fewer labeled samples each. Experimental results indicate that MoleVers achieves competitive performance on them, suggesting its robustness in both low-data settings.

**Strengths:**

- **Valuable Research Question**: The research question addressed in this paper focuses on an area in the field that has lacked sufficient attention and remains underexplored. The authors provide a feasible solution for effective property prediction in low-data scenarios.

- **Benchmark Contribution**: The MPPW benchmark is a valuable addition to the field, as it reflects real-world scenarios where labeled molecular data is often scarce.

- **Strong Empirical Results**: MoleVers achieves consistently high performance across various assays, demonstrating its generalizability and effectiveness in low-data regimes.

**Weaknesses:**

**Weaknesses:**

- My primary concern with this paper is the lack of technical novelty. Although the authors propose a two-stage pretraining approach, each pretraining stage employs widely used methods, such as MAP and dynamic denoising. Thus, in terms of technical contribution, this work may lean more toward engineering improvements.

- Some experimental setups and procedures lack sufficient motivation and explanation:

   1. In Section 3.2, the authors mention selecting HOMO, LUMO, and Dipole Moment as auxiliary tasks. However, as quantum properties, these may be less beneficial for ADMET tasks, raising doubts about the validity of the assumption in line 226.

   2. In line 298, the authors use a subset of the stage 1 pretraining dataset for auxiliary task labeling and further pretraining in stage 2. This implementation detail lacks motivation—why was a subset of the stage 1 pretraining dataset chosen for stage 2?

   3. In Section 5.4, the authors point out that the benefit from stage 2 pretraining might depend on the nature of the downstream task, which heightens concerns over the choice of task properties in stage 2.

   4. In line 425 of Section 5.6, the authors explore the impact of pretraining dataset diversity by filtering molecules based on certain elements. However, do these filtered datasets differ in size? This could also contribute to performance differences, so it would be preferable if the authors controlled for dataset size in this analysis.

- The authors’ evaluation in high-data settings is insufficiently comprehensive. They selected only three datasets (QM7, QM8, QM9), which are closely related to the auxiliary tasks. I recommend a more diverse downstream task evaluation on MoleculeNet or TDC datasets.

**Questions:**

Please refer to the questions in Weaknesses.

---

> ### Author Response · Authors · 2024-11-19
>
> **Each pretraining stage employs widely used methods such as MAP and denoising. Thus, in terms of technical contribution, this work may lean more toward engineering improvements.**
>
> We are in full agreement that masked atom prediction (MAP) and denoising tasks, both used in our first pretraining stage, have been widely adopted in various works. However, the technical novelty of our first pretraining stage lies in the dynamic denoising and the branching encoder, which decouples the denoising and MAP tasks. This novel design fixes a significant problem in denoising pretraining, where it was believed that large noise scales (e.g., those leading to larger than **1 Å** atom displacement) typically hurt the downstream performance [1-4]. With our model, we demonstrate that using larger noise scales (up to **10 Å**) actually improves the downstream performance—a new insight for denoising-based molecular pretraining.
>
> We also note that our technical contributions go beyond the pretraining strategy. Specifically, we:
> - Propose a new benchmark (Section 4) that reflects the conditions in the real-world. Here, the datasets contain 50 or less training samples with experimentally-validated properties. We believe our benchmark is more realistic compared with QMs or MoleculeNet, which use datasets with hundreds to hundreds of thousand training samples. For reference, only around 0.37% of assays in the ChemBL database have 100 or more labeled molecules.
> - Standardize the pretraining dataset in our benchmark to limit the influence of pretraining dataset quality to the downstream performance. The standardization is important to ensure fairness when comparing multiple pretraining methods, as pretraining dataset quality can significantly affect the downstream performance (Section 5.6)
>  - Provide extensive discussions on the effects of (1) noise scales, (2) pretraining dataset size, (3) pretraining dataset diversity.n the effects of (1) noise scales, (2) pretraining dataset size, (3) pretraining dataset diversity.
>
>
> **Quantum properties may be less beneficial for ADMET tasks, raising doubts about the validity of the assumption in line 226, especially with the results in Section 5.4.**
>
> Our hypothesis is based on the fact that all molecular properties are a function of their 3D structures (Line 232). Therefore, the pretraining properties do not have to be strongly-correlated with the downstream properties for the supervised pretraining to be beneficial. This is demonstrated in Table 1, as our model achieves good performance across the diverse type of properties in MPPW.
> Note that the benefits of stage 2 might be reduced, but do not entirely vanish, on properties with weaker correlation to the pretraining properties (Section 5.4.). This is supported by the results in Table 2 row 3, which  still consistently outperform the results in Table 2 row 1.
>
> **Why was a subset of the stage 1 pretraining dataset chosen for stage 2?**
>
> One advantage of splitting the pretraining into two stages is that we can use any unlabeled dataset in stage 1 and any labeled dataset in stage 2. Using a subset of the dataset used in stage 1 pretraining is a practical choice as we already have the data ready in the pipeline.
>
> **Do these filtered datasets in Section 5.6 differ in size? This could also contribute to performance differences, so it would be preferable if the authors controlled for dataset size in this analysis.**
>
> Thank you for pointing this out! The dataset size could indeed affect the downstream performance. That is why we fixed the training set size to 100K for each row in Table 6. We have added this detail to the revised paper.
>
> **The authors’ evaluation in high-data settings is insufficiently comprehensive. I recommend a more diverse downstream task evaluation on MoleculeNet or TDC datasets.**
>
> Thank you for the suggestion! We have conducted additional experiments on regression tasks in the MoleculeNet benchmark (high-data regime), and the results are shown in the table below. Combined with results in Table 7, we can see that our method outperforms state-of-the-art models in high-data regimes. Note that we have trained the baselines from scratch using the GDB17 dataset. This is to ensure that any performance gains are the results of the pretraining strategy, not the pretraining dataset quality.
>
> It is important to note that our work primarily focuses on real-world scenarios where labeled training data often consists of 50 molecules or fewer. Therefore, the results on our MPPW benchmark (small-data regime) should be considered as the core evaluation, while results from QMs and MoleculeNet are included only for supplementary comparisons.

---

> > ### Author Response · Authors · 2024-11-19
> >
> > Table A. Results on MoleculeNet (regression tasks). We report the RMSE of each method.
> > | Dataset   | Ours (pretrained with GDB17) | UniMol (pretrained with GDB17) | Mol-AE (pretrained with GDB17) |
> > |-----------|-------------------------------|--------------------------------|--------------------------------|
> > | ESOL      | **0.8036**                        | 0.8497                       | 0.9524                         |
> > | LIPO      | **0.6354**                        | 0.6465                         | 0.6965                         |
> > | Freesolv  | 1.5944                       | **1.5769**                         | 2.5861                         |

---

> ### Comment · Reviewer_kQgY · 2024-11-24
>
> Thank you for the authors' response. Most of my concerns have been addressed; however, I still have reservations regarding the novelty of the work. I will make a final decision on whether to adjust my score after further discussion with the other reviewers.

---

### Official Review · Reviewer_yqCY · 2024-11-04

**Soundness:** 2
**Presentation:** 3
**Contribution:** 2
**Rating:** 5
**Confidence:** 3

**Summary:**

This paper designs a two-stage pre-training framework for molecular property prediction in the wild. The authors claim that previous methods are not good at generalizing to the testing cases with few labeled samples provided. The first stage of the proposed method additionally equips denoising objectives as branching encoder learning compared to the previous method. The second stage is training the model via weakly supervised learning with coarse labels (also through computation tools). Experimental results on a new benchmark dataset demonstrate the proposed method's effectiveness.

**Strengths:**

(1) The presentation of this paper is quite clear. The major idea is quite straightforward. It can be expected that better denoising frameworks and weakly supervised learning can be effective for enhancing few-shot molecular property prediction performance.

(2) Comprehensive experiments have been covered in this work. And new large-scale benchmark dataset has been introduced for evaluation in this work.

**Weaknesses:**

(1) The algorithmic contribution appears somewhat limited. Firstly, both the denoising pretraining strategy and masking strategy have been explored in previous works. Therefore, this study primarily concentrates on the interaction between these two modules during the pretraining stage, which lacks novelty. In the second stage, the effectiveness of weakly supervised learning is not surprising, provided the computed coarse labels are essentially accurate. Overall, the efficacy of the proposed strategies has been largely validated by previous works, which somewhat diminishes the significance of this study.

(2) It appears that the research works related to few-shot molecular property prediction have been overlooked in the discussion of related works. Interestingly, the experiments conducted in this study primarily focused on the few-shot molecular property prediction task, yet this line of research has not been addressed in the related works section.

(3) Given that the benchmark dataset introduced in this work is novel, it is not entirely convincing that the proposed method consistently outperforms previous approaches. The number of baseline methods covered is relatively small compared to the total number of molecular pretraining methods. It seems that some molecular denoising pretraining frameworks, which are crucial for validating the effectiveness and novelty of the proposed method, have not been included in the comparison.

**Questions:**

(1) Could the authors elaborate more on the novelty of the first stage of pre-training? What are the key novel points that distinguish the proposed method from previous approaches?

(2) Could authors have more discussions over related works about the few-shot molecular property prediction? There is also a series of works working on this problem. Considering this work is actually tackling the same problem, it is appropriate to have a few more discussions and even experimental comparisons over the previous approaches.

---

> ### Author Response · Authors · 2024-11-19
>
> **Both the denoising pretraining strategy and masking strategy have been explored in previous works. The efficacy of the proposed strategies has been largely validated by previous works.**
>
> We are in full agreement that masked atom prediction (MAP) and denoising tasks, both used in our first pretraining stage, have been widely adopted in various works. However, the technical novelty of our first pretraining stage lies in the dynamic denoising and the branching encoder, which decouples the denoising and MAP tasks. This novel design fixes a significant problem in denoising pretraining, where it was believed that large noise scales (e.g., those leading to larger than **1 Å** atom displacement) typically hurt the downstream performance [1-4]. With our model, we demonstrate that using larger noise scales (up to **10 Å**) actually improves the downstream performance—a new insight for denoising-based molecular pretraining that has never been validated in previous works.
>
> We also note that our technical contributions go beyond the pretraining strategy. Specifically, we:
> - Propose a new benchmark (Section 4) that reflects the conditions in the real-world. Here, the datasets contain 50 or less training samples with experimentally-validated properties. We believe our benchmark is more realistic compared with QMs or MoleculeNet, which use datasets with hundreds to hundreds of thousand training samples. For reference, only around 0.37% of assays in the ChemBL database have 100 or more labeled molecules.
> - Standardize the pretraining dataset in our benchmark to limit the influence of pretraining dataset quality to the downstream performance. The standardization is important to ensure fairness when comparing multiple pretraining methods, as pretraining dataset quality can significantly affect the downstream performance (Section 5.6)
> - Provide extensive discussions on the effects of (1) noise scales, (2) pretraining dataset size, (3) pretraining dataset diversity.
>
> **Could authors have more discussions over related works about the few-shot molecular property prediction?**
>
> Thank you for the feedback! Although related, our work does not fall into the traditional few-shot learning framework. Prior studies in this area [7-9] typically formulate the few-shot prediction as an N-way K-shot classification problem, where N classes of molecules are sampled from a dataset, each with K examples [7]. This formulation is not applicable to regression tasks, which are the focus of the proposed MPPW benchmark. Therefore, we choose baseline models that follow a pretraining-finetuning paradigm. We have clarified this in the revised related work section.
>
>
> **It seems that some molecular denoising pretraining frameworks, which are crucial for validating the effectiveness and novelty of the proposed method, have not been included in the comparison.**
>
> Due to space limitation, we choose to include UniMol and Mol-AE in Table 1 as they are the latest, best performing methods using MAP and denoising pretraining. We also include GNN-based methods, GraphMVP and Mole-BERT, as additional baselines. Does the reviewer have suggestions on which additional baselines are necessary to be added? We will try to add it during the discussion phase.

---

> > ### Comment · Reviewer_yqCY · 2024-11-26
> > **Reply to Authors**
> >
> > Apologies for the late reply. Having carefully reviewed the rebuttal, this reviewer would like to share the following observations:
> >
> > (1) Concerning the matter of contributions, it seems to this reviewer that the current paper presents a variety of unique contributions from different perspectives. However, these contributions appear somewhat disjointed, and each section lacks sufficient depth. This reviewer would strongly advise the authors to focus on a single, significant contribution and delve deeper into it.
> >
> > (2) The proposed method employs weakly supervised learning, which could potentially enhance the performance of the method. However, this reviewer feels that the use of weakly supervised learning is not fully exploited in this work. Previous studies have also explored this strategy for improving molecular potential estimation, and it would be beneficial for this work to take these studies into account.
> >
> > (3) The current baseline methods are adequate.
> >
> > As a result, this reviewer will retain the current rating score for the time being, and may contemplate modifications after further discussions with other reviewers.
> >
> > Best Regards.

---

### Author Response · Authors · 2024-11-19
**General Response**

We thank the reviewers for their insightful comments. We are pleased that all reviewers recognize the value of our research question of property prediction in the small-data regime and appreciate the comprehensiveness of our experimental evaluation (yqCY, kQgY). We are also encouraged by the positive reception of the proposed two-stage approach and dynamic denoising, which were noted as innovative and inspiring (StiN, BLw3). Here, we address the recurring comments regarding our contributions.

The contributions of this work can be summarized as follows:
- **The branching encoder and dynamic denoising**. While masked atom prediction and denoising tasks have been employed in various works, denoising with larger noise scales (e.g., larger than **1 Å**) is previously thought to be detrimental to the downstream prediction performance [1-4]. With our novel design, we demonstrated the contrary that using larger noise levels (up to **10 Å**) in the first pretraining stage can actually improve the downstream performance.

- **The two-stage pretraining approach**. We demonstrate that performing a supervised pretraining following the self-supervised pretraining helps with prediction performance, even if the pretraining and downstream properties are not strongly-correlated. Moreover, splitting the pretraining into two stages allows our strategy to be used in a plug and play manner. Thus, we can use any unlabeled datasets (stage1) and labeled datasets (stage 2) to train any models (Appendix A2).

- **A new benchmark for molecular property prediction that reflects real-world conditions**. Existing benchmarks, such as [6], usually provide 600 to 130K labeled molecules. This number of labeled samples is unrealistic; only around 0.37% of assays in the ChemBL database have 100 or more labels. The proposed MPPW benchmark addresses this gap by using experimentally-validated datasets that mostly contain 50 or less training samples.


Reference

[1] Zhou, Gengmo, et al. "Uni-mol: A universal 3d molecular representation learning framework." (2023).

[2] Yang, Junwei, et al. "MOL-AE: Auto-Encoder Based Molecular Representation Learning With 3D Cloze Test Objective." (2024).

[3] Zaidi, Sheheryar, et al. "Pre-training via denoising for molecular property prediction." (2022).

[4] Ni, Yuyan, et al. "Pre-training with fractional denoising to enhance molecular property prediction." (2024).

[5] Luo, Shengjie, et al. "One transformer can understand both 2d & 3d molecular data." (2023).

[6] Wu, Zhenqin, et al. "MoleculeNet: a benchmark for molecular machine learning." (2018).

[7] Ju, Wei, et al. "Few-shot molecular property prediction via hierarchically structured learning on relation graphs." (2023)

[8] Guo, Zhichun, et al. "Few-shot graph learning for molecular property prediction." (2021)

[9] Wang, Yaqing, et al. "Property-aware relation networks for few-shot molecular property prediction." (2021)

[10] Lee, Juho, et al. "Set transformer: A framework for attention-based permutation-invariant neural networks." International conference on machine learning. (2019)

---

### Meta-Review · Area_Chair_fHdw · 2024-12-22

**Metareview:**

(a) Summary: The paper proposes MoleVers, a two-stage pretrained model for molecular property prediction. In the first stage, it uses masked atom prediction and dynamic denoising with a branching encoder on large unlabeled datasets. The second stage employs supervised pretraining with auxiliary labels from inexpensive computational methods. The model is evaluated on a new benchmark (MPPW) with 22 datasets having mostly 50 or fewer training labels.

(b) Strengths:

   - The research addresses an important problem of property prediction with limited experimentally-validated data.
   - The introduction of the MPPW benchmark is valuable as it reflects real-world conditions with scarce labeled data better than existing benchmarks.
  - The paper presents comprehensive experimental evaluations.

(c) Weaknesses:
- The technical novelty of the pretraining strategy is questioned, as masked atom prediction and denoising tasks have been used before. The key innovation of dynamic denoising and branching encoder needs further justification.
- Some experimental details lack sufficient motivation and explanation, such as the choice of auxiliary tasks and the use of a subset of the stage 1 dataset in stage 2.
- The paper could have more in-depth discussions on related works, especially in the area of few-shot molecular property prediction.

(d) Rejection Reasons:
- The lack of clear and significant technical novelty in the proposed pretraining approach is a major concern. Although the authors attempt to highlight the dynamic denoising and branching encoder, the overall contribution seems incremental.
- The insufficient motivation and explanation for certain experimental setups make it difficult to fully assess the validity and significance of the proposed method.
- The limited discussion on related works and the potential overlap with existing research directions reduce the impact and uniqueness of the paper.

**Additional Comments On Reviewer Discussion:**

During the rebuttal period, reviewers raised several points. One key point was the novelty of the first stage of pre-training. The authors addressed this by emphasizing the dynamic denoising and branching encoder, which they claim fixes a problem in previous denoising pretraining and improves downstream performance. However, some reviewers remained unconvinced about the novelty. Regarding the choice of auxiliary tasks in the second stage, the authors explained their hypothesis that pretraining properties don't need to be strongly correlated with downstream properties, but this did not fully satisfy the reviewers. The authors also provided additional experiments and explanations for other concerns, such as the impact of branching and dynamic noise. However, the overall consensus among reviewers was that the paper still had significant weaknesses in terms of novelty and experimental justifications. In the final decision, these unresolved issues weighed heavily, leading to the rejection of the paper as it did not meet the standards for acceptance in terms of making a substantial and novel contribution to the field.

---

### Decision · Program_Chairs · 2025-01-22

Reject